# An Attempt to Predict Changes in Heart Rate Variability in the Training Intensification Process among Cyclists

**DOI:** 10.3390/ijerph18147636

**Published:** 2021-07-18

**Authors:** Paulina Hebisz, Rafał Hebisz, Agnieszka Jastrzębska

**Affiliations:** Department of Physiology and Biochemistry, University School of Physical Education in Wrocław, 35 I.J. Paderewski Avenue, 51-612 Wrocław, Poland; rafalhebisz@poczta.fm (R.H.); agnieszka.jastrzebska@awf.wroc.pl (A.J.)

**Keywords:** heart rate variability, high-intensity interval training, cardiorespiratory fitness

## Abstract

Individual changes in resting heart rate variability (HRV) parameters were assessed in seven Polish cyclists during a training process consisting of: a six-week period (P1) of predominantly low- and moderate-intensity training (L-MIT) and a six-week period (P2) where the proportion of high-intensity interval training (HIT) increased. Daily recorded HRV parameters included high-frequency spectral power (HF), square root of the mean squared difference between successive normal-to-normal RR intervals (RMSSD), and standard deviation of normal-to-normal RR intervals (SDNN). In each training microcycle, the average values of HF_av_, RMSSD_av_, and SDNN_av_ were calculated individually for each participant. In three cyclists, HF was higher in P2 compared to P1, whereas in one cyclist, HF was higher in P1 than in P2. Each of these four cyclists presented an individual correlation between the average daily duration HIT effort in training microcycles (HIT_av_) and HF_av_. Cyclists with low baseline values of HRV parameters showed increased activity of the parasympathetic nervous system, while in the cyclist with high baseline values of HRV parameters, an opposite change was observed. In conclusion, changes in resting HRV parameters between period P1 and P2 can be individualised. In the investigated group, it was possible to predict how HRV would change as a result of training intensification on the basis of HRV baseline values.

## 1. Introduction

Training intensification comprises changes in the intensity, duration, and frequency of training sessions in a training process lasting from several weeks to even several months [1]. According to Solli et al. [2], training intensification is a classic periodization model for endurance athletes. On the other hand, Smith [3] indicates that the training periodization is not only planning the distribution of training loads, but also planning the frequency of races, tests assessing the level of athletes’ efficiency, and planning regeneration periods. Among endurance athletes, low- and moderate-intensity training (L-MIT) accounts for 80% of total training load, and 20% is high-intensity interval training (HIT) [1]. Training intensification consists in reducing the duration and frequency of L-MIT training sessions and increasing the duration and frequency of HIT training [1].

Literature distinguishes two variations of interval training, one involving exercise performed at maximal intensity (sprint interval training) with the other at high intensity above the lactate/ventilatory threshold or with an intensity at 95–125% maximal aerobic power [4,5,6]. The interval training protocol includes multiple repetitions of exercise, often lasting from a few seconds to several minutes, separated by recovery periods of varying duration [4,5,6,7]. However, the effects of training intensification on cardiorespiratory and performance changes are not clear.

According to some studies, training intensification is a process that positively influences the development of cardiorespiratory efficiency [4,8], as measured by the level of maximal oxygen uptake (VO_2_max) [9]. On the other hand, studies on cross-country skiers by Evertsen et al. [10] did not show any significant differences in physiological and performance changes between moderate-intensity and high-intensity training groups. Gaskill et al. [11], as a result of a two-year research project, describe athletes achieving improvements in physiological test results and race performances by implementing traditional training loads, with a predominance of L-MIT training, and a second group of athletes with no performance improvement resulting from a traditional approach. However, the application of training intensification in this group achieved the expected results. The above reports indicate that the issue of training intensification should be approached individually, as it is beneficial only for some athletes.

The effects of training intensification are assessed on the basis of physiological and performance changes [1,11]. Up to now, researchers have mainly been focused on adaptive changes to the training process in general, and thus individuals may be missed. The phenomena of individual reactions and adaptive changes dependent on specific training are known facts. Therefore, it seems reasonable to look for factors that could characterize individuals. This will allow one to more accurately predict the development of cardiorespiratory fitness in response to a several-week training intensification programme consisting of L-MIT and HIT efforts.

A popular method to evaluate cardiorespiratory fitness is an analysis of autonomic nervous system activity based on the records of resting [12,13,14] or restitution [15] sinus heart rate variability (HRV). A measurement of HRV is often used to assess training load [16] and post-exercise fatigue in athletes [17,18,19]. Moreover, changes in HRV parameters are analysed in relation to the level of aerobic capacity, as measured by the value of maximal oxygen uptake [9]. Ueno et al. [14] and Oliveira et al. [20] observed that obtaining high aerobic capacity was accompanied by an increase in HRV parameters identifying vagus nerve activation. Moreover, Botek et al. [21] found that maintaining high resting high-frequency spectral power (HF), total spectral power (T), and the square root of the mean squared difference between successive values of normal-to-normal RR intervals (RMSSD) in the training process allows for the effective development of aerobic capacity. Høydal [22] and Garber et al. [23] have shown that participants with a high VO_2_max level struggle to achieve further improvement in VO_2_max through training, as compared to individuals with a low VO_2_max level. This raises the question of whether, among athletes with high values of HRV parameters, it will be more difficult to improve them in the training process because, as it has been shown, VO_2_max correlates with HRV parameters [14,20]. Many athletes divide the training process into a period of predominant moderate-intensity efforts and a period of increased high-intensity efforts [2,24]. However, available literature lacks information on whether one can predict resting HRV changes in the training intensification process. 

Therefore, the aim of this study was to evaluate individual changes in resting HRV parameters in cyclists during a training process wherein, after a period of predominant L-MIT training, the proportion of HIT training increased. It was assumed that, in cyclists characterised by low (as compared with other observed individuals) baseline resting HF and RMSSD, training intensification would result in relatively large changes in these parameters, while in cyclists with high (as compared with other participants) baseline resting HF and RMSSD, training intensification would result in relatively small changes in these parameters.

## 2. Materials and Methods

### 2.1. Participants

Seven Polish mountain-bike cyclists (four men: S1, S3, S5, S6; and three women: S2, S4, S7) participated in this study. Prior to the experiment, each cyclist had at least three years of training experience and had participated in cycling races at the national level. Based on the maximal oxygen uptake, six cyclists can be rated at aerobic fitness level, while the S3 cyclist, at an elite level, in accordance with the criteria proposed by Figueira et al. [25] and Joyner and Coyle [26]. Table 1 presents the physiological and anthropometric characteristics of the studied cyclists and their experience in practicing cycling.

The study design was approved by the local Ethics Committee of the University School of Physical Education (Consent number: 39/2019), and all procedures were performed in accordance with the Declaration of Helsinki. Written informed consent was obtained from the participants and their legal guardians after the study details, procedures, risks, and benefits had been explained.

### 2.2. Experimental Design 

The experiment was preceded by a 2-week period of weekly training loads reduced to 3 sessions of 45 min each with an intensity of 70–85% of the second ventilatory threshold (VT2) power. The investigation consisted in observing the effects of the training process carried out for 12 weeks and was developed on the basis of the classic periodisation model [2,27]. The study was divided into 2 periods, each lasting 6 weeks. In the first period (P1), L-MIT trainings predominated. In the second period (P2), the number and duration of L-MIT trainings were reduced (as compared to the last week of P1), while the number and duration of HIT trainings were increased. In a few cases, the periods were shortened by several days as the participants presented with health problems. In the event of an infection, no extension of any period was allowed, owing to commitments to participate in competitions. In each case, data analysis was stopped 10 days before the first race of the competition period. A flowchart showing the course of the experiment is shown in Figure 1.

In each training period, 4-day training microcycles were implemented. Each microcycle ended with a day of rest (4th day of the microcycle). In P1, the microcycles consisted of the following cycling trainings:L-MIT trainings at a level of 70–85% power measured at VT2 and at 65–75% of maximal heart rate (HRmax); their duration equalled 2–4 h (Training 1—T1);trainings with exercises requiring high pedalling frequency, at a heart rate of 65–75% HRmax—repeated efforts with a pedalling frequency increased by 15–25 RPM compared to the individually preferred rhythm (determined on the basis of constant-intensity training observations) (Training 2—T2);training sessions with repeated exercises of high intensity (above 150% of maximal aerobic power (Pmax), determined in the incremental exercise test) lasting 15–20 s (Training 3—T3);trainings consisting of resistance exercises (e.g., semi-squats) alternated with cycling exercises of high pedalling frequency (increased by 15–25 RPM compared to the individually preferred rhythm), at a heart rate of 65–75% HRmax (Training 4—T4).

Each microcycle included 2 L-MIT trainings and 1 training selected from among the others mentioned above. In P2, the following cycling trainings were implemented:L-MIT trainings at an intensity of 70–85% power measured at VT2 and at a heart rate of 65–75% HRmax; their duration equalled 2–3.5 h (Training 5—T5);HIT interval trainings involving repeated efforts at 130–160% Pmax lasting 25–50 s (Training 6—T6);HIT trainings comprised of repeated efforts at 105–120% Pmax lasting 1–2 min (Training 7—T7);HIT trainings consisting of repeated efforts at 90–100% Pmax lasting 3–6 min (Training 8—T8);trainings composed of resistance exercises (e.g., semi-squats) (Training 9—T9).

Each microcycle included 2 HIT trainings selected from among those mentioned above and 1 L-MIT training. In addition, in each period, L-MIT running trainings (at 60–70% maximum heart rate, determined in the incremental exercise test) and plyometric trainings (based on jumping ability exercises) were implemented. In each period, it was possible to extend the training microcycle by 1 day, during which an additional L-MIT training was performed, provided that: (1) the participants evaluated their general feeling as good during the whole microcycle (the evaluated was made on the basis of a conversation with the cyclists and the assessment of the ability to perform the given training loads—when the cyclist reported that he/she could easily perform subsequent training sessions, his well-being was assessed as good); (2) the LF/HF ratio determined during the microcycle did not assume values exceeding the individual average value by more than 100% (such a criterion was adopted based on the observation of the LF/HF index values in the work with cyclists, the results of which are described in this manuscript—such a change usually resulted in a decrease in the ability to perform training sessions); (3) no decrease in power was observed during training compared with previous training microcycles. In each period, it was possible to shorten the training microcycle by 1 day, provided that: (1) the participants evaluated their general feeling as bad during the whole microcycle; (2) the LF/HF ratio determined during the microcycle assumed values exceeding the individual average value by more than 100%; (3) a decrease in power was observed during training compared with previous training microcycles.

During both training periods, it was assumed that the cyclists (performing L-MIT training) should try to keep the heart rate close to that recorded in the first L-MIT training performed during P1 and P2.

Information showing the workload of each cyclist is provided in Table 2.

### 2.3. Exercise Test

Immediately before each training period, an incremental exercise test was performed on a Lode Excalibur Sport cycle ergometer (Lode BV, Groningen, the Netherlands), calibrated before commencement of the study. The test started with a 50 W load; every 3 min, the load was increased by 35 and 50 W (for women and men, respectively) until refusal due to exhaustion. If the participant was unable to exercise for the entire 3 min at the last test load, 0.19 and 0.28 W (for women and men, respectively) was subtracted from the obtained final maximal power for each missed second [4,28,29]. In this way, Pmax was calculated.

During the test, respiratory parameters were recorded. The subjects wore a mask connected to a Quark breathing gas analyser (Cosmed, Milan, Italy). The gas analyser was calibrated before the test, via connection with a gas cylinder containing a reference gas mixture consisting of carbon dioxide (5%), oxygen (16%), and nitrogen (79%). Respiratory parameters were measured breath by breath and then averaged in 30 s intervals. VO_2_max was indicated on the basis of the recorded data, if the subject met at least two of the following three criteria: (1) 90% of age-predicted maximal heart rate (220—age); (2) respiratory exchange ratio >1.15; and (3) lactate concentration >10 mmol·l^−1^ [30]. Arterialized capillary blood was drawn 3 min after the test’s conclusion to assay lactate concentration by a Lactate Scout (SensLab, Leipzig, Germany). VT2 was indicated with the method of Beaver et al. [31], at the point preceding the second non-linear increase in VE∙VO_2_^−1^ or VE∙VCO_2_^−1^ equivalent, and the power output at VT2 was then determined.

The previous week, at the beginning of the study and throughout the experiment, the time interval between heartbeats (RR) was recorded with a V800 chest strap and heart rate monitor (Polar Electro Oy, Kempele, Finland). Each recording was performed on a daily basis for 10 min immediately after waking up, in a supine position. A 5-min segment starting with the 30th second of the recording was analysed. The days on which the athletes had an infection and the first training microcycle from restarting training after the infection were excluded from the analysis. For each recording, the following parameters were calculated: high-frequency spectral power (HF), low-frequency spectral power (LF), total spectral power (T), the square root of the mean squared difference between successive normal-to-normal RR intervals (RMSSD), standard deviation of normal-to-normal RR intervals (SDNN), and the mean normal-to-normal RR intervals (RRNN). The calculations were performed with Kubios HRV Standard software (Kubios Oy, Kuopio, Finland) using the fast Fourier transformation. A low artifact correction threshold was used when performing the analysis.

In each training microcycle, the daily duration of L-MIT effort (performed with a power below VT2) and the daily duration of HIT effort (performed with a power above 90% Pmax) was recorded. These calculations included only cycling efforts. The remaining efforts were not considered in the data analysis.

For each participant, average power (P_av_) and average heart rate (HR_av_) were recorded during 60 min of cycling, immediately after a warm-up, in the L-MIT training implemented in the first microcycle of P1, in the last microcycle of P1 and in the last microcycle of P2. PowerTap P1 power meters (PowerTap, Madison, WI, USA) were used, the reliability and validity of which were determined by Wright et al. [32]. The data analysis was carried out via the flow.polar.com Internet platform, which is used to generate files with a record of heart rate [33,34]. This platform automatically calculated the average values of the parameters for the selected part of the training. The obtained data served to calculate the P_av_/HR_av_ ratio.

### 2.4. Statistical Calculations and Analyses

Statistica 13.1 software (StatSoft Inc. Tulsa, OK, USA) was used for statistical calculations. The arithmetic mean and standard deviation values of all analysed HRV parameters, L-MIT effort duration, and HIT effort duration were established for P1 and P2 individually for each cyclist. Student’s *t*-test served to individually analyse the probability of difference between P1 and P2 for the mean values of HRV parameters, L-MIT effort duration, and HIT effort duration. A level of probability below 0.05 was considered statistically significant. Based on the arithmetic mean and standard deviations values, Cohen’s D values were calculated.

Average values of all analysed HRV parameters for the previous week at the beginning of the study (HF_b_, LF_b_, RMSSD_b_, SDNN_b_, RRNN_b_) and average values of all analysed HRV parameters for each implemented training microcycle during experiment (HF_av_, LF_av_, RMSSD_av_, SDNN_av_, RRNN_av_) were calculated individually for each cyclist. Moreover, the average daily L-MIT effort duration (L-MIT_av_) and the average daily HIT effort duration (HIT_av_) in each training microcycle were determined individually for each cyclist.

The Pearson correlation coefficient between the average values of each analysed HRV parameters in the microcycles (HF_av_, LF_av_, RMSSD_av_, SDNN_av_, RRNN_av_) and average training loads in the microcycles (L-MIT_av_, HIT_av_) was indicated individually for each participant.

For the whole study group, Pearson correlation coefficient was calculated between the average values of each HRV parameter in the week preceding the experiment commencement (HF_b_, LF_b_, RMSSD_b_, SDNN_b_, RRNN_b_) and the strength of the individual Pearson correlations between the average values of HRV parameters in the microcycles (HF_av_, LF_av_, RMSSD_av_, SDNN_av_, RRNN_av_) and the average training loads in the microcycles (L-MIT_av_, HIT_av_). For each of the Pearson analyses, a probability level below 0.05 was considered statistically significant.

To assess changes in P_av_ and HR_av_ and P_av_/HR_av_ ratio, a one-way analysis of variance with repeated measures and a post-hoc Scheffe test were performed. The probability level below 0.05 was considered statistically significant.

## 3. Results

HF differed between P1 and P2 only in the S1, S2, S3, and S7 cyclists. In the S1, S2, and S3 participants, HF was higher in P2, whereas in S7, HF was higher in P1. Moreover, in the S1 and S2 participants, RMSSD was higher in P2 than in P1. The S7 participant presented RMSSD and SDNN values higher in P1 than in P2. In individual cases, differences in LF, SDNN, and RRNN between P1 and P2 were also observed. In the S4 and S5 cyclists, no difference was noted in any HRV parameter between P1 and P2 (Table 3).

The P_av_/HR_av_ ratio increased in the last L-MIT training as compared to the first L-MIT training during P1 by more than 0.10 W∙BPM^−1^ in the S1, S3, S5, and S6 cyclists. The P_av_/HR_av_ ratio increased in the last L-MIT training of P2 as compared to the last L-MIT training of P1 by more than 0.10 W∙BPM^−1^ in the S1, S2, S3, and S7 cyclists. The analysis of variance showed statistically significant main effects for the repeated measures P_av_ (7.73; *p* = 0.007; η2 = 0.56) and P_av_/HR_av_ (F = 10.03; *p* = 0.002; η2 = 0.63) (Table 4).

A statistically significant correlation between HIT_av_ and HF_av_ was observed. The correlation was positive in the S1, S2, and S3 cyclists and negative in the S7 cyclist. Correlations of the required level of probability were detected between HIT_av_, RMSSD_av_, and SDNN_av_. These correlations were positive in the S1 and S2 cyclists and negative in the S7 cyclist. In the case of S5, a positive correlation of the required level of statistical probability was identified between HIT_av_ and LF_av_ only. The S4 and S6 cyclists did not present correlations of the required level of statistical probability for HIT_av_. No athlete demonstrated statistically significant correlations for L-MIT_av_ (Table 5 and Figure 2).

It was found that the levels of HF_b_, LF_b_, RMSSD_b_, and SDNN_b_ measured in the week preceding the experiment were statistically significantly correlated with the strength of the relationship between HIT_av_ and HF_av_, HIT_av_ and RMSSD_av_ and HIT_av_ and SDNN_av_ (Table 6 and Figure 3).

## 4. Discussion

In this study, it was shown that, in three cyclists, HF was higher in period P2 than in P1, whilst in the case of one cyclist, HF was higher in period P1 than in P2. In the remaining three cyclists, there were no significant changes in HF. Interestingly, in all four cyclists who had changes in HF between P1 and P2, the Pav/HRav ratio from the last L-MIT training of period P2 was higher than the last L-MIT training of P1. An analysis of the available literature shows that resting heart rate and HRV values are related to aerobic capacity and cardiorespiratory fitness among non-training individuals [13,35]. According to Bellenger et al. [36], an increase in exercise capacity during the training process is associated with an improvement in HRV parameters, such as RMSSD or HF. Sandercock et al. [37] proved in their meta-analysis that a long-term training process based on aerobic effort leads to an increase in resting HF values; Macor et al. [38] reported that competitive cyclists present higher resting HF values than non-training individuals. However, publications describing the impact that intensifying training has on HRV parameters provide inconsistent results. Pichot et al. [39] demonstrated that, over a period of three weeks, when intensive training accounted for 30% of the total load, the value of HF decreased, although the subject of the study was training that provoked fatigue accumulation. Schneider et al. [40] implied that among well-trained athletes, the natural logarithm of RMSSD did not change after several HIT trainings. Similarly, Daniłowicz-Szymanowicz et al. [41] did not observe changes in HF, RMSSD, or SDNN as a result of a two-month training intensification among trained runners. Raczak et al. [42] compared two training periods of four months each in trained runners. The first period was characterised by a predominance of moderate-intensity training. In the second period, the proportion of high-intensity trainings increased. The RMSSD, SDNN, and LF parameters turned out to be significantly higher in the second period than in the preceding period. The results of the present study indicate that changes in HRV parameters due to intensifying the training process can be individualised among well-trained cyclists.

Lamberts et al. [43] demonstrated that the power output level at submaximal heart rate was a good measure of exercise capacity in cycling. In our study, the above-mentioned HRV parameters improved in P2 among the cyclists S1, S2, and S3. In the same period, the P_av_/HR_av_ ratio improved in these cyclists. However, an improvement in the P_av_/HR_av_ ratio was also observed in cyclist S7, who presented decreased HF, SDNN, and RMSSD in P2. It therefore seems that controlling the training so as to achieve the highest possible parameters reflecting vagus nerve activity is not the sole condition for improving power output at submaximal heart rate.

It would be reasonable to indicate the variables that could serve to predict the effects of training intensification on HRV parameters in a situation where P2 was associated with an increase in vagus nerve activity in three cyclists and no relationship, or an opposite one, was observed in the remaining cases. The potential to make such predictions seems valuable for coaches and athletes in the context of studies that found a relationship between HRV parameters that identify vagus nerve activity and the ability of athletes to perform intense physical effort [20,35]. An attempt to predict the impact of training on HRV was made by Chalencon et al. [44], but their model was based on the observation of a training process implemented earlier. Our intention, in turn, was to try to determine whether it was possible to assess the efficacy of training intensification in terms of its impact on HRV on the basis of initial HRV measurements (taken before the training process). Intensifying training is an element of periodisation and constitutes a common practice among endurance athletes during the several weeks preceding important competitions [2,24,45]. In our research, we applied a training intensification similar to the classic model. This concept assumes that a period of predominantly L-MIT trainings is followed by a period with a decreasing proportion of L-MIT and a simultaneous increasing proportion of HIT [2]. Our findings (Table 5) imply that baseline HRV (HF, LF, RMSSD, and SDNN) can be a factor that determines the strength and direction of the relationship between HIT_av_ and HRV variables (HF_av_, RMSSD_av_, and SDNN_av_). This information supplements our previous research in which we proved that the effects of periods of training intensification through the use of HIT and sprint interval training could be predicted by an analysis of the training volume in a previous training process [28,29], as well as peak pulmonary minute ventilation in incremental tests and the restitution RMSSD value after a moderate-intensity warm-up [28].

## 5. Conclusions

The presented study results indicate that changes in resting HRV parameters between a period of L-MIT training predominance and a period of increased HIT training proportion can be individualised. Daily records of resting HRV and training loads can allow one to determine whether an increase in high-intensity loads affects changes in HRV parameters in individual cases. In the investigated group, it was possible to predict how HRV would change as a result of training intensification on the basis of the baseline values of HRV parameters, such as HF, LF, RMSSD, and SDNN. Their increase was observed in cyclists with low baseline values of these parameters, and their decrease was found in the cyclist with high baseline values. It also seems that training intensification can also lead to power output improvement at a submaximal heart rate in individuals other than those with an increase in parameters reflecting parasympathetic nervous system activation.

## Figures and Tables

**Figure 1 ijerph-18-07636-f001:**
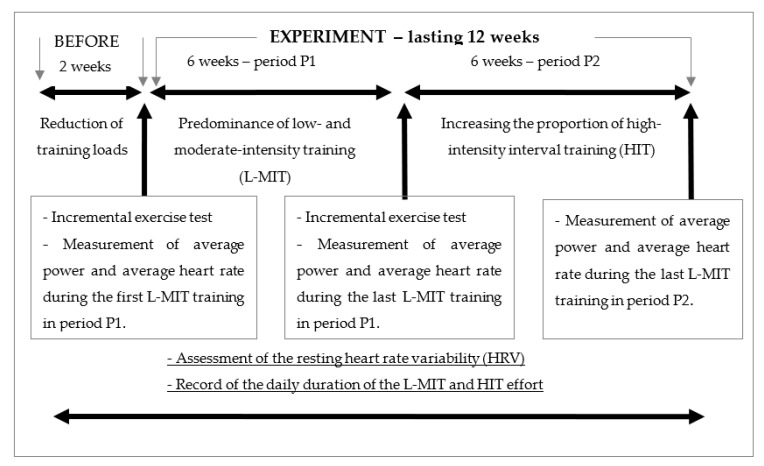
Flowchart of the study design.

**Figure 2 ijerph-18-07636-f002:**
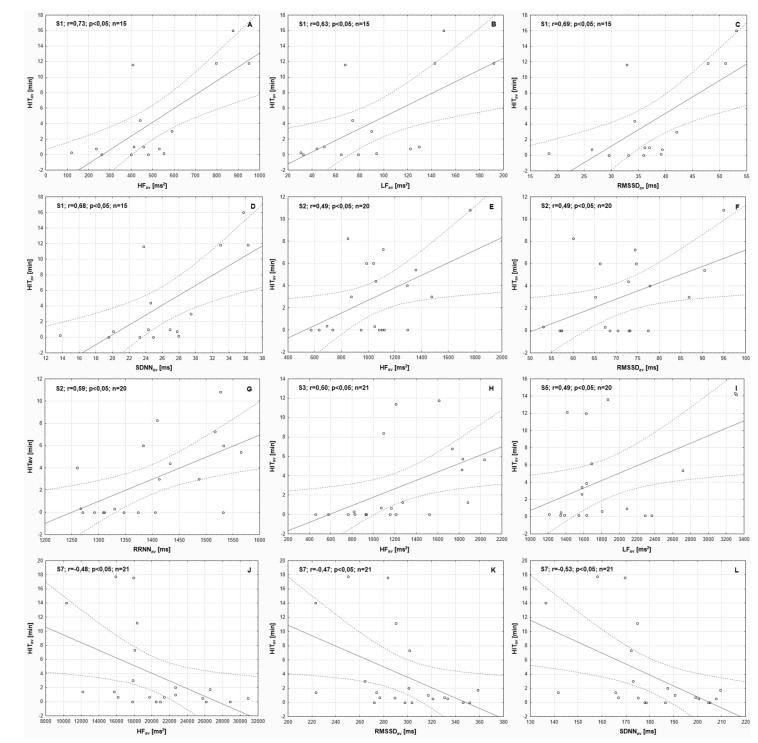
Graphical representation of the statistically significant Pearson’s correlations (indicated in Table 5) between daily duration of high-intensity training—averaged values for training microcycles (HIT_av_) and average parameters of sinus heart rate variability recorded in the subsequent training microcycles throughout the experiment (totals for the first and second period). S1, S2, etc.—subsequent participants; *r*—correlation coefficient; *p*—the adopted level of statistical significance; *n*—number of training microcycles performed. **ParA**—shows the correlation between HIT_av_ and average value of high-frequency spectral power, recorded in the subsequent training microcycles throughout the experiment (totals for the first and second period) (HF_av_)—for participant S1; **Part B**—shows the correlation between HIT_av_ and average value of low-frequency spectral power, recorded in the subsequent training microcycles throughout the experiment (LF_av_)—for participant S1; **Part C**—shows the correlation between HIT_av_ and average value of square root of the mean squared difference between successive normal-to-normal RR intervals, recorded in the subsequent training microcycles throughout the experiment (RMSSD_av_)—for participant S1; **Part D**—shows the correlation between HIT_av_ and average value of standard deviation of normal-to-normal RR intervals, recorded in the subsequent training microcycles throughout the experiment (SDNN_av_)—for participant S1; **Part E**—shows the correlation between HIT_av_ and HF_av_—for participant S2; **Part F**—shows the correlation between HIT_av_ and RMSSD_av_—for participant S2; **Part G**—shows the correlation between HIT_av_ and average value of mean normal-to-normal RR intervals, recorded in the subsequent training microcycles throughout the experiment (RRNN_av_)—for participant S2; **Part H**—shows the correlation between HIT_av_ and HF_av_—for participant S3; **Part I**—shows the correlation between HIT_av_ and LF_av_—for participant S5; **Part J**—shows the correlation between HIT_av_ and HF_av_—for participant S7; **Part K**—shows the correlation between HIT_av_ and RMSSD_av_— for participant S7; **Part L**—shows the correlation between HIT_av_ and SDNN_av_—for participant S7.

**Figure 3 ijerph-18-07636-f003:**
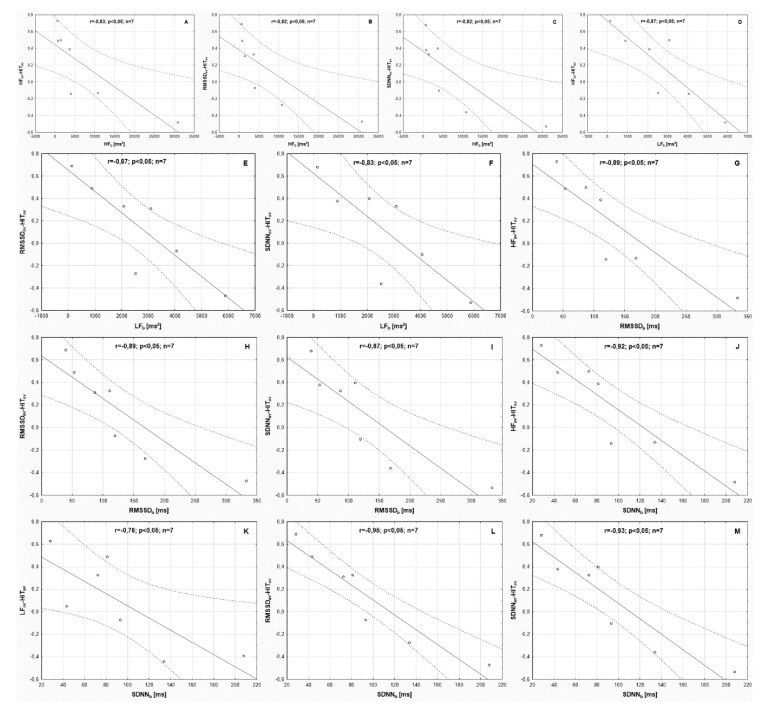
Graphical representation of the statistically significant Pearson’s correlations (indicated in Table 6) between the strength of the relationship between the daily duration of high-intensity training—averaged values for training microcycles (HIT_av_) and parameters of sinus heart rate variability calculated for the whole experiment period and parameters of sinus heart rate variability recorded in the week preceding the experiment. *r*—correlation coefficient; *p*—the adopted level of statistical significance; *n*—number of participants. **Part A**—shows the correlation between the strength of the relationship between the average value of high-frequency spectral power, recorded in the subsequent training microcycles throughout the experiment (totals for the first and second period) (HF_av_) and HIT_av_, as well as high-frequency spectral power expressed as the average of measurements performed in the week preceding the experiment (HF_b_); **Part B**—shows the correlation between the strength of the relationship between the average value of the square root of the mean squared difference between successive normal-to-normal RR intervals, recorded in the subsequent training microcycles throughout the experiment (RMSSD_av_) and HIT_av_, as well as HF_b_; **Part C**—shows the correlation between the strength of the relationship between the average value of the standard deviation of normal-to-normal RR intervals (SDNN_av_) and HIT_av_, as well as HF_b_; **Part D**—shows the correlation between the strength of the relationship between HF_av_ and HIT_av_, as well as low-frequency spectral power expressed as the average of measurements performed in the week preceding the experiment (LF_b_); **Part E**—shows the correlation between the strength of the relationship between RMSSD_av_ and HIT_av_, as well as LF_b_; **Part F**—shows the correlation between the strength of the relationship between SDNN_av_ and HIT_av_, as well as LF_b_; **Part G**—shows the correlation between the strength of the relationship between HF_av_ and HIT_av_, as well as the square root of the mean squared difference between successive normal-to-normal RR intervals expressed as the average of measurements performed in the week preceding the experiment (RMSSD_b_); **Part H**—shows the correlation between the strength of the relationship between RMSSD_av_ and HIT_av_, as well as RMSSD_b_; **Part I**—shows the correlation between the strength of the relationship between SDNN_av_ and HIT_av_, as well as RMSSD_b_; **Part J**—shows the correlation between the strength of the relationship between HF_av_ and HIT_av_, as well as standard deviation of normal-to-normal RR intervals expressed as the average of measurements performed in the week preceding the experiment (SDNN_b_); **Part K**—shows the correlation between the strength of the relationship between the average value of low-frequency spectral power, recorded in the subsequent training microcycles throughout the experiment (LF_av_) and HIT_av_, as well as SDNN_b_; **Part L**—shows the correlation between the strength of the relationship between RMSSD_av_ and HIT_av_, as well as SDNN_b_; **Part M**—shows the correlation between the strength of the relationship between SDNN_av_ and HIT_av_, as well as SDNN_b_.

**Table 1 ijerph-18-07636-t001:** Individual characteristics of the study participants based on the incremental exercise test, HRV analyses performed before the experiment and competitive status.

Variables	S1	S2	S3	S4	S5	S6	S7
Age [year]	21	18	24	17	20	22	20
Body mass [kg]	75.4	55.8	73.2	49.3	62.2	70.9	54.7
Body height [m]	1.83	1.71	1.81	1.62	1.71	1.86	1.63
VO_2_max [ml∙min^−1^∙kg^−1^]	66.6	58.1	77.0	58.0	62.4	68.7	60.4
Pmax_1_ [W∙kg^−1^]	5.97	4.70	6.52	4.56	5.66	5.97	5.34
Pmax_1_ [W]	450	262	477	225	352	423	292
Pmax_2_ [W]	460	275	465	240	344	418	310
VT2_1_ [W]	305	205	320	140	225	305	205
VT2_2_ [W]	330	190	335	145	220	305	215
HF_b_ [ms^2^]	546	689	1381	3922	3597	10,746	30,865
LF_b_ [ms^2^]	137	890	3082	4060	2086	2526	5881
RMSSD_b_ [ms]	39	53.1	86.1	119.3	111	168.3	333.4
SDNN_b_ [ms]	27.9	43.2	72.3	93.1	81.1	133.7	207.7
RRNN_b_ [ms]	1006	1267	1537	1140	1318	1197	1398
Experience [y]	7	4	9	3	6	7	5

HRV—heart rate variability; S1—first participant of the study (a man cyclist); S2—second participant of the study (a woman cyclist); S3—third participant of the study (a man cyclist); S4—fourth participant of the study (a woman cyclist); S5—fifth participant of the study (a man cyclist); S6—sixth participant of the study (a man cyclist); S7—seventh participant of the study (a woman cyclist); VO_2_max—maximal oxygen uptake; Pmax_1_—maximal aerobic power measured before period P1; Pmax_2_—maximal aerobic power measured before period P2; VT2_1_—power at the second ventilatory threshold measured before period P1; VT2_2_—power at the second ventilatory threshold measured before period P2; HF_b_—high-frequency spectral power; LF_b_—low-frequency spectral power; RMSSD_b_—square root of the mean squared difference between successive normal-to-normal RR intervals; SDNN_b_—standard deviation of normal-to-normal RR intervals; RRNN_b_—mean normal-to-normal RR intervals; _b_—each HRV parameter was expressed as the average of measurements performed in the week preceding the commencement of the experiment; Experience—experience in practicing cycling and competitive status presented in years.

**Table 2 ijerph-18-07636-t002:** Individual load in cycling training.

Cycling Training		S1	S2	S3	S4	S5	S6	S7
**T1**	t (min)	120–240	120–210	120–225	120–150	150–230	120–195	150–240
	P (W)	215–260	145–175	225–270	100–120	160–190	215–260	145–175
	Rep	1	1	1	1	1	1	1
**T2**	t (min)	10–20	10–15	10–15	5–10	10–15	10–15	10–20
	RPM	103–113	101–111	97–107	106–116	100–110	98–108	106–116
	P (W)	215–260	145–175	225–270	100–120	160–190	215–260	145–175
	Rep	2–5	2–4	2–5	1–4	2–5	2–5	2–5
**T3**	t (s)	15	15	15	15	15	15	15
	P (W)	≥675	≥393	≥715	≥337	≥528	≥634	≥438
	Rep	1–4	1–4	1–4	1–4	1–4	1–4	1–4
**T4**	t (min)	3	2	3	1.5	3	2	3
	RPM	103–113	101–111	97–107	106–116	100–110	98–108	106–116
	P (W)	290–320	190–220	305–335	125–155	210–240	290–320	190–220
	Rep	3–7	3–6	3–7	2–5	3–7	3–7	3–7
**T5**	t (min)	120–195	120–160	120–195	120–135	135–195	120–180	140–210
	P (W)	230–280	130–160	235–285	100–125	155–185	215–260	150–180
	Rep	1	1	1	1	1	1	1
**T6**	t (s)	40–50	30–40	40–50	25–30	40–50	40–50	40–50
	P (W)	600–650	380–420	605–655	355–385	450–500	540–590	400–440
	Rep	8–16	6–12	8–20	4–10	8–16	8–12	8–12
**T7**	t (s)	60–120	60–90	60–120	60–80	60–90	60–120	60–120
	P (W)	480–520	290–330	485–525	250–290	370–410	440–480	325–365
	Rep	5–10	3–7	5–8	3–6	4–8	3–7	5–10
**T8**	t (min)	4–6	3–4	4–6	3–3.5	4–6	3–5	5–6
	P (W)	410–450	250–275	415–455	215–240	310–345	375–415	280–310
	Rep	4–8	3–6	4–8	2–4	4–8	3–6	4–8

S1, S2, etc.—subsequent participants; T1, T2, etc.—subsequent types of cycling training; t—duration of training efforts; P—power achieved during training efforts; RPM—pedalling frequency for selected training efforts; Rep—the number of repetitions in one training session.

**Table 3 ijerph-18-07636-t003:** Changes in the parameters of sinus heart rate variability and daily training loads in the first (P1) and second (P2) training period. Results presented as individual cases for S1–S7 participants and as the arithmetic mean value for the entire group of cyclists.

Participants	HF [ms^2^]	LF [ms^2^]	RMSSD [ms]	SDNN [ms]	RRNN [ms]	HIT [min]	L-MIT [min]
**S1**							
P1 (*n* = 31)	367.7 ±217.8	64.1 ±57.1	31.9 ±9.8	22.5 ±6.7	978.8 ±53.0	0.3 ±0.9	94.2 ±77.7
P2 (*n* = 30)	644.1 ±401.0 **	119.3 ±71.4 **	42.3 ±13.9 **	29.8 ±8.9 **	991.1 ±44.7	8.0 ±13.3 **	82.0 ±58.7
*t*	−3.36	−3.34	−3.37	−3.62	−0.98	−3.24	0.69
D	0.89	0.86	0.88	0.94	0.25	1.08	0.18
**S2**							
P1 (*n* = 35)	927.7 ±439.1	743.5 ±687.2	64.9 ±15.8	44.7 ±11.6	1315.6 ±81.1	0.1 ±0.3	81.7 ±71.2
P2 (*n* = 43)	1184.6 ±566.5 *	795.4 ±544.9	76.7 ±17.6 **	51.4 ±15.6 *	1457.6 ±108.8 **	6.0 ±9.1 **	79.2 ±68.0
*t*	−2.20	−0.37	−3.09	−2.11	−6.62	−3.81	−0.52
D	0.51	0.08	0.71	0.49	1.50	1.26	0.04
**S3**							
P1 (*n* = 43)	1023.8 ±524.3	1928.3 ±1048.5	76.6 ±16.3	60.4 ±13.1	1588.5 ±75.0	0.1 ±0.3	95.2 ±82.0
P2 (*n* = 40)	1522.1 ±816.5 **	2177.8 ±1723.2	81.0 ±13.4	64.9 ±14.1	1511.4 ±89.1 **	6.1 ±10.1 **	84.8 ±67.2
*t*	−3.33	−0.80	−1.33	−1.52	4.27	−3.97	0.63
D	0.74	0.18	0.30	0.33	0.94	1.15	0.14
**S4**							
P1 (*n* = 36)	4101.4 ±1733.2	2272.9 ±1146.8	106.7 ±25.8	81.8 ±16.2	1043.6 ±83.1	0.6 ±1.7	53.9 ±56.3
P2 (*n* = 39)	4476.8 ±2349.7	2398.5 ±1111.0	109.3 ±25.1	85.7 ±15.4	1027.2 ±72.8	3.3 ±4.8 **	46.5 ±50.0
*t*	−0.78	−0.48	−0.45	−1.05	0.91	−3.21	0.61
D	0.18	0.11	0.10	0.25	0.21	0.83	0.14
**S5**							
P1 (*n* = 42)	4330.1 ±1140.4	1719.9 ±789.5	122.4 ±14.6	85.1 ±9.6	1347.8 ±108.7	0.7 ±2.1	95.4 ±82.9
P2 (*n* = 35)	4699.2 ±1593.9	1997.4 ±1204.3	127.3 ±17.6	88.1 ±13.3	1339.7 ±85.5	8.51 ±12.9 **	71.1 ±60.3
*t*	−1.17	−1.20	−1.33	−1.11	0.35	−3.85	1.44
D	0.27	0.28	0.30	0.26	0.08	1.04	0.34
**S6**							
P1 (*n* = 34)	7568.7 ±4036.0	4012.3 ±3715.5	148.1 ±40.3	118.4 ±32.7	1277.0 ±106.1	0.1 ±0.3	84.7 ±78.9
P2 (*n* = 42)	6529.8 ±4667.4	2747.4 ±2800.9	130.3 ±46.9	101.3 ±35.5 *	1295.8 ±74.2	3.3 ±6.7 **	78.6 ±67.4
*t*	1.01	1.67	1.73	2.13	−0.90	−2.88	0.36
D	0.24	0.39	0.41	0.50	0.21	0.91	0.08
**S7**							
P1 (*n* = 43)	23,603.9 ±5928.9	6291.4 ±2258.2	317.2 ±41.6	193.1 ±19.4	1483.3 ±91.6	0.6 ±1.7	103.1 ±92.0
P2 (*n* = 35)	15,732.1 ±4225.4 **	4639.7 ±2137.8 **	264.2 ±40.1 **	161.6 ±21.4 **	1434.6 ±95.2 *	8.7 ±16.3 **	96.7 ±77.8
*t*	6.60	3.29	5.69	6.82	2.29	−3.26	0.33
D	1.55	0.75	1.30	1.54	0.52	0.90	0.08
**Mean**							
P1	6403.9 ±8471.5	2542.7 ±2637.1	129.5 ±93.6	89.7 ±55.8	1310.9 ±219.7	0.4 ±1.3	87.5 ±79.1
P2	4773.9 ±5352.5	2092.2 ±2062.3	116.9 ±68.8	82.2 ±42.3	1312.2 ±207.2	5.9 ±10.7	74.9 ±65.3

HF—high-frequency spectral power; LF—low-frequency spectral power; RMSSD—square root of the mean squared difference between successive normal-to-normal RR intervals; SDNN—standard deviation of normal-to-normal RR intervals; RRNN—mean normal-to-normal RR intervals; HIT—daily duration of high-intensity training; L-MIT—daily duration of low- and moderate-intensity training; S1, S2, etc.—subsequent participants; P1—first training period; P2—second training period; *n*—number of heart rate variability records performed; *t*—Student’s *t*-test value; data is presented as mean ± standard deviation; D—the value of D-Cohen’s statistic; Mean—the value of the arithmetic mean ± standard deviation for the entire group of cyclists; * *p* < 0.05, significant difference between the P1 and P2 value; ** *p* < 0.01, significant difference between the P1 and P2 value.

**Table 4 ijerph-18-07636-t004:** Power and heart rate during the investigated low- and moderate-intensity trainings.

Participants	Variables	First L-MIT in P1	Last L-MIT in P1	Last L-MIT in P2
S1	P_av_ [W]	212	232	275
	HR_av_ [BPM]	142	141	143
	P_av_/HR_av_ [W∙BPM^−1^]	1.49	1.65	1.92
S2	P_av_ [W]	152	155	175
	HR_av_ [BPM]	149	146	150
	P_av_/HR_av_ [W∙BPM^−1^]	1.02	1.06	1.17
S3	P_av_ [W]	252	265	284
	HR_av_ [BPM]	151	142	142
	P_av_/HR_av_ [W∙BPM^−1^]	1.67	1.87	2.00
S4	P_av_ [W]	153	141	160
	HR_av_ [BPM]	145	141	152
	P_av_/HR_av_ [W∙BPM^−1^]	1.05	1.00	1.05
S5	P_av_ [W]	208	227	234
	HR_av_ [BPM]	157	155	155
	P_av_/HR_av_ [W∙BPM^−1^]	1.32	1.46	1.51
S6	P_av_ [W]	237	271	251
	HR_av_ [BPM]	157	160	152
	P_av_/HR_av_ [W∙BPM^−1^]	1.51	1.69	1.65
S7	P_av_ [W]	160	164	176
	HR_av_ [BPM]	142	146	141
	P_av_/HR_av_ [W∙BPM^−1^]	1.13	1.12	1.25
Mean	P_av_ [W]	196.3 ± 41.4	207.9 ± 53.8	222.1 ± 51.3 *
	HR_av_ [BPM]	149.0 ± 6.4	147.3 ± 7.4	147.9 ± 5.7
	P_av_/HR_av_ [W∙BPM^−1^]	1.31 ± 0.25	1.41 ± 0.35	1.51 ± 0.37 *

L-MIT—low- and moderate-intensity training; P1—first training period; P2—second training period; S1, S2, etc.—subsequent participants; P_av_—average power during 60 min of cycling in the indicated training; HR_av_—average heart rate during 60 min of cycling in the indicated training; Mean—the value of the arithmetic mean ± standard deviation for the entire group of cyclists; *—*p* < 0.05 vs. first L-MIT in P1.

**Table 5 ijerph-18-07636-t005:** Strength of the Pearson correlation between average training loads and average parameters of sinus heart rate variability recorded in the subsequent training microcycles throughout the experiment (totals for the first and second period).

Participants Training load	HF_av_ [ms^2^]	LF_av_ [ms^2^]	RMSSD_av_ [ms]	SDNN_av_ [ms]	RRNN_av_ [ms]
S1 (*n* = 16)					
HIT_av_ [min]	0.73 *	0.63 *	0.69 *	0.68 *	0.26
L-MIT_av_ [min]	0.04	0.23	0.00	0.04	–0.45
S2 (*n* = 20)					
HIT_av_ [min]	0.49 *	0.05	0.49 *	0.38	0.59 *
L-MIT_av_ [min]	–0.11	0.11	–0.21	0.03	–0.39
S3 (*n* = 21)					
HIT_av_ [min]	0.50 *	0.33	0.31	0.33	–0.42
L-MIT_av_ [min]	–0.11	0.18	0.06	0.24	–0.23
S4 (*n* = 23)					
HIT_av_ [min]	–0.14	–0.07	–0.07	–0.10	–0.18
L-MIT_av_ [min]	0.18	–0.15	0.10	0.14	0.01
S5 (*n* = 20)					
HIT_av_ [min]	0.39	0.49 *	0.33	0.40	–0.30
L-MIT_av_ [min]	0.09	–0.18	0.13	0.08	0.21
S6 (*n* = 19)					
HIT_av_ [min]	–0.13	–0.44	–0.27	–0.36	0.14
L-MIT_av_ [min]	0.19	–0.11	0.28	0.23	–0.38
S7 (*n* = 21)					
HIT_av_ [min]	–0.48 *	–0.39	–0.47 *	–0.53 *	–0.27
L-MIT_av_ [min]	0.04	–0.14	0.14	0.07	0.38

HF—high-frequency spectral power; LF—low-frequency spectral power; RMSSD—square root of the mean squared difference between successive normal-to-normal RR intervals; SDNN—standard deviation of normal-to-normal RR intervals; RRNN—mean normal-to-normal RR intervals; S1, S2, etc.—subsequent participants; *n*—number of training microcycles performed; HIT_av_—daily duration of high-intensity training, averaged values for training microcycles; L-MIT_av_—daily duration of low- and moderate-intensity training, averaged values for training microcycles; * *p* < 0.05.

**Table 6 ijerph-18-07636-t006:** Strength of the Pearson correlation between parameters of sinus heart rate variability recorded in the week preceding the experiment and the strength of the relationship between the training loads and parameters of sinus heart rate variability calculated for the whole experiment period.

Variables	HF_av_-HIT_av_	LF_av_-HIT_av_	RMSSD_av_-HIT_av_	SDNN_av_-HIT_av_	RRNN_av_-HIT_av_
HF_b_ [ms^2^]	–0.83 *	–0.69	–0.82 *	–0.82 *	–0.31
LF_b_ [ms^2^]	–0.87 *	–0.64	–0.87 *	–0.83 *	–0.67
RMSSD_b_ [ms]	–0.89 *	–0.72	–0.89 *	–0.87 *	–0.42
SDNN_b_ [ms]	–0.92 *	–0.78 *	–0.95 *	–0.93 *	–0.44
RRNN_b_ [ms]	–0.15	–0.14	–0.25	–0.21	–0.57

HF_av_-HIT_av_—Pearson correlation between the average value of high-frequency spectral power and average high-intensity load in the training microcycles; LF_av_-HIT_av_—Pearson correlation between the average value of low-frequency spectral power and average high-intensity load in the training microcycles; RMSSD_av_-HIT_av_—Pearson correlation between the average value of the square root of the mean squared difference between successive normal-to-normal RR intervals and average high-intensity load in the training microcycles; SDNN_av_-HIT_av_—Pearson correlation between the average value of the standard deviation of normal-to-normal RR intervals and average high-intensity load in the training microcycles; RRNN_av_-HIT_av_—Pearson correlation between the average value of the normal-to-normal RR intervals and average high-intensity load in the training microcycles; HF_b_—high-frequency spectral power; LF_b_—low-frequency spectral power; RMSSD_b_—square root of the mean squared difference between successive normal-to-normal RR intervals; SDNN_b_—standard deviation of normal-to-normal RR intervals; RRNN_b_—mean normal-to-normal RR intervals; _b_—each HRV parameter was expressed as the average of measurements performed in the week preceding the experiment; * *p* < 0.05.

## Data Availability

Not applicable.

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
