# Peer review of "An Attempt to Predict Changes in Heart Rate Variability in the Training Intensification Process among Cyclists"

_ijerph, 2021, doi:10.3390/ijerph18147636_

Round 1

Reviewer 1 Report

Changes made substantially improve the paper.
Congratulations        

Author Response

Reviewer 1

Comments and Suggestions for Authors

Changes made substantially improve the paper.
Congratulations      

- Thank you very much for reviewing our manuscript and your positive feedback.

Reviewer 2 Report

Review

Manuscript number: ijerph-1282424

Title: “An Attempt to Predict Changes in Heart Rate Variability in the 2 Training Intensification Process Among Cyclists”

 This study is experimental study to evaluate individual changes in resting HRV 73 parameters in cyclists during a training process in which, after a period of predominant 74 LIT training, the proportion of HIT training increased. It is well described article with proper method and conclusion. It is acceptable

Title

Good

Abstract

Good

Methods

Good.

Results

Good.

Discussion

Good.

Conclusion

Good.

References

Good.

Author Response

Reviewer 2

Comments and Suggestions for Authors

This study is experimental study to evaluate individual changes in resting HRV parameters in cyclists during a training process in which, after a period of predominant LIT training, the proportion of HIT training increased. It is well described article with proper method and conclusion. It is acceptable

- Thank you very much for reviewing our manuscript and your positive feedback.

Reviewer 3 Report

First of all, I am not English native speaker, so I hope you understand me well.

The authors do a good job with this study. The discussion section could be refined somewhat. The other sections are better and need minor changes.

Further comments and suggestions:

Line 29. Low and moderate-intensity training should have another abbreviation. L-MIT may be? Or low (LIT) and moderate-intensity training (MIT).

Line 38. What is not clear? The effects of training intensification on what parameter?

Line 51. I think that “training intensification” is not the best words to define your approach. You are focus on the changes of training intensity distribution or number of high-intensity training through the season (in different phases). This is what periodization is about, right?

Line 70. High-intensity loads? High loads can be high volume of training.

Line 75. Your assumption is not based on previous evidence during the introduction section. Please, provide more information above.

Line 83. Please modify to: “Seven Polish mountain-bike cyclists….participated in this study”.

Table 1. You must concrete the meaning of S1, S2…at the bottom of the table.

Line 106. Infections? Maybe change to a health problem or similar.

Figure 1. I don’t know what you want to say with “tested first LIT in P1, P2…”.

Line 148. He/she

Line 171. This criterion needs a reference.

Line 174. The subjects wore…

Line 179. Please, provide the VO2max selection criteria (RER, blood lactate, HR…).

Line 183. Change to “The previous week at the beginning of the study…”

Line 184. Change “cardiofrequencimeter” to” chest strap and heart rate monitor”.

Line 218. Change the “experiment commencement” as before.

In the discussion section, you use present and past in the sentences. Maybe, it is better use present or past consistently,

Line 362. It better not to use “If” to start a paragraph.

Author Response

Reviewer 3

Comments and Suggestions for Authors

First of all, I am not English native speaker, so I hope you understand me well.

The authors do a good job with this study. The discussion section could be refined somewhat. The other sections are better and need minor changes.

- Thank you very much for your review and suggestions.

Further comments and suggestions:

Line 29. Low and moderate-intensity training should have another abbreviation. L-MIT may be? Or low (LIT) and moderate-intensity training (MIT).

- Indeed, the abbreviation LIT is not appropriate, it has been changed to L-MIT.

Line 38. What is not clear? The effects of training intensification on what parameter?

- We had cardiorespiratory and performance changes in mind. The sentence has been corrected: "However, the effects of training intensification on cardiorespiratory and performance changes are not clear"

Line 51. I think that “training intensification” is not the best words to define your approach. You are focus on the changes of training intensity distribution or number of high-intensity training through the season (in different phases). This is what periodization is about, right?

- That's right, periodization is about changing of training intensity distribution or number of high-intensity training through the season. However, on the other hand, the training periodization is not only planning the distribution of training loads, but also planning the frequency of races, tests assessing the level of athletes' efficiency, and planning regeneration periods. The concept of training periodization has a broader meaning than training intensification. Therefore, we would prefer to use the term "intensification" in the presented article.

- Such information was added at the beginning of the Introduction section.

Line 70. High-intensity loads? High loads can be high volume of training.

- Right, that's not an appropriate term, it has been changed to „efforts”.

Line 75. Your assumption is not based on previous evidence during the introduction section. Please, provide more information above.

- As suggested, more information has been added in the Introduction section. I hope this is a sufficient basis for our assumption.

Line 83. Please modify to: “Seven Polish mountain-bike cyclists….participated in this study”.

- The sentence has been changed.

Table 1. You must concrete the meaning of S1, S2…at the bottom of the table.

- According to the suggestion, the meaning of the indicated abbreviations has been explained.

Line 106. Infections? Maybe change to a health problem or similar.

- "infections" has been changed to "a health problem"

Figure 1. I don’t know what you want to say with “tested first LIT in P1, P2…”.

- The indicated part of the text in Figure 1 has been corrected for a more detailed description:

"Measurement of average power and average heart rate during the first L-MIT training in period P1. ......"

Line 148. He/she

- Done

Line 171. This criterion needs a reference.

- The reference has been added.

Line 174. The subjects wore…

- The sentence has been changed.

Line 179. Please, provide the VO2max selection criteria (RER, blood lactate, HR…).

- The criteria for the indication of VO2max have been added in the Material and Methods section.

Line 183. Change to “The previous week at the beginning of the study…”

- The sentence has been changed.

Line 184. Change “cardiofrequencimeter” to” chest strap and heart rate monitor”.

- Done

Line 218. Change the “experiment commencement” as before.

- The sentence has been changed. 

In the discussion section, you use present and past in the sentences. Maybe, it is better use present or past consistently,

- As suggested, the discussion section has been corrected by a native speaker.

Line 362. It better not to use “If” to start a paragraph.

- The sentence has been changed

This manuscript is a resubmission of an earlier submission. The following is a list of the peer review reports and author responses from that submission.

Round 1

Reviewer 1 Report

INTRODUCTION

With two very long paragraphs the approach to the target becomes very dense. Moreover intensification of training is referenced but nothing is said about different types of HIT.

Warr-di Piero D, Valverde-Esteve T, Redondo-Castán JC, Pablos-Abella C, Sánchez-Alarcos Díaz-Pintado JV (2018). Effects of work-interval duration and sport specificity on blood lactate concentration, heart rate and perceptual responses during high intensity interval training. PLoS ONE 13(7): e0200690. https://doi.org/10.1371/journal.pone.0200690

MATERIAL AND METHODS

Participants

It would be important to know the competitive level of the subjects. Judging by the Pmáx (with quite low values) it does not seem that they are high-competitive-level subjects. This aspect should be clearly reflected as it could be a limitation to be highlighted in the discussion.

In Table 1, values of some variables should be compared with reference values. So for example, It should be possible to identify the range for VO2 (from poor to excellent) and whether a is acceptable that HFb range from 546 to 30865 or RMSDD range from 39 to 333..)

Experimental design

It would be interesting display a flowchart for recruitment and testing

In line 88 “… of the classic Matveyev’s periodization model [19]” I consider that the reference is not correct because the cited study does not justify Matveyev's model. In this sense, in my opinion the theoretical models of training planning (Matveiev, Verjosanski…) should not be cited in this context:

Kiely, J (2012). Periodization Paradigms in the 21st Century: Evidence-Led or Tradition-Driven? International Journal of Sports Physiology and Performance, 7, 242-250.

As an alternative, I propose that the experimental design be based on the application of regular loads (applicable for athletes of medium-low competitive level). Which is precisely why I believe it would be necessary to express the workload of each athlete.

Training by periods is very well detailed however, in my opinion, subsequent explanations have no validity because the real load has not been expressed. Real workouts cannot be range from 70 to 85% or from 130 to 160%. it would be necessary to know the real load carried out by each subject

In lines 167-168 “…data analysis was carried out with the flow.polar.com Internet platform,” I believe that it should give validity and reliability to this platform, demonstrating in which other scientific studies have been used or explained and how they have validated it.

Statistical calculations and analyses

The goal is to make a prediction, so I think the individual results only add noise (it should be attached as an annex)

At least the effect size is required in order to measure the reliability and validity of the statistic

Results

The results should show the way to “Predict Changes in Heart Rate Variability” so, as mentioned above the individual results only add noise

Obviousness, as expressed in lines 195-196 (“In all cyclists, the daily HIT training volume was statistically significantly higher in P2 compared with P1 (Tables 2a and 2b)”) should be avoided.

DISCUSION

Needs a thorough review. The results of the study (based on the hypothesis formulated initially) should be clearly stated

Very long paragraphs that do not discuss in an orderly manner the aspects investigated.

Author Response

INTRODUCTION

With two very long paragraphs the approach to the target becomes very dense. Moreover intensification of training is referenced but nothing is said about different types of HIT.

Warr-di Piero D, Valverde-Esteve T, Redondo-Castán JC, Pablos-Abella C, Sánchez-Alarcos Díaz-Pintado JV (2018). Effects of work-interval duration and sport specificity on blood lactate concentration, heart rate and perceptual responses during high intensity interval training. PLoS ONE 13(7): e0200690. https://doi.org/10.1371/journal.pone.0200690

- Thank you very much for your valuable comments. The introduction has been improved, now describes the issue of training intensification, different types of HIT training and the use of HRV measurement in the training process.

MATERIAL AND METHODS

Participants

It would be important to know the competitive level of the subjects. Judging by the Pmáx (with quite low values) it does not seem that they are high-competitive-level subjects. This aspect should be clearly reflected as it could be a limitation to be highlighted in the discussion.

In Table 1, values of some variables should be compared with reference values. So for example, It should be possible to identify the range for VO2 (from poor to excellent) and whether a is acceptable that HFb range from 546 to 30865 or RMSDD range from 39 to 333..)

- Information describing the competitive level of the participants has been added in the Material and Methods section.

Experimental design

It would be interesting display a flowchart for recruitment and testing

- A flowchart has been attached to the Materials and Methods section.

In line 88 “… of the classic Matveyev’s periodization model [19]” I consider that the reference is not correct because the cited study does not justify Matveyev's model. In this sense, in my opinion the theoretical models of training planning (Matveiev, Verjosanski…) should not be cited in this context:

Kiely, J (2012). Periodization Paradigms in the 21st Century: Evidence-Led or Tradition-Driven? International Journal of Sports Physiology and Performance, 7, 242-250.

- The indicated sentence has been changed.

As an alternative, I propose that the experimental design be based on the application of regular loads (applicable for athletes of medium-low competitive level). Which is precisely why I believe it would be necessary to express the workload of each athlete.

- As suggested, the workload for each athlete is shown in the current version of the manuscript. This information has been added to the Materials and Methods section in Table 2.

Training by periods is very well detailed however, in my opinion, subsequent explanations have no validity because the real load has not been expressed. Real workouts cannot be range from 70 to 85% or from 130 to 160%. it would be necessary to know the real load carried out by each subject

- This information has been added to the Materials and Methods section in Table 2.

In lines 167-168 “…data analysis was carried out with the flow.polar.com Internet platform,” I believe that it should give validity and reliability to this platform, demonstrating in which other scientific studies have been used or explained and how they have validated it.

- In the indicated fragment of the manuscript, information on the use of this platform in other scientific studies has been addend.

Statistical calculations and analyses

The goal is to make a prediction, so I think the individual results only add noise (it should be attached as an annex)

- Showing individual reactions different is the sense of this manuscript. In the current version of the manuscript, mean values for the entire study group of cyclists have been added - but only as additional data - because they do not reflect individual variation.

At least the effect size is required in order to measure the reliability and validity of the statistic

- Added the value of D-Cohen's statistic to the student's t-test results.

Results

The results should show the way to “Predict Changes in Heart Rate Variability” so, as mentioned above the individual results only add noise

- The sense and purpose of this manuscript is to show that the prediction of HRV changes is individual. So the presentation of individual results is crucial, we take the average values only as an addition.

Obviousness, as expressed in lines 195-196 (“In all cyclists, the daily HIT training volume was statistically significantly higher in P2 compared with P1 (Tables 2a and 2b)”) should be avoided.

- The indicated sentence has been deleted.

DISCUSION

Needs a thorough review. The results of the study (based on the hypothesis formulated initially) should be clearly stated

Very long paragraphs that do not discuss in an orderly manner the aspects investigated.

- As suggested in the Discussion section, changes have been made, I hope they will be sufficient.

Reviewer 2 Report

The study examined the individual changes on HRV after a period of low and a period of high intensity training. The approach to express the individual variation is import and useful as it applied in the present study. Several limitations should be discussed, for example: i) training history before the commencement of the study, ii) the internal training load was not estimated. An effort should be applied to connect the introduction with the findings and discussion. A large part of introduction is connecting VO2max with HRV, however, this not obvious in the results and discussion. Finally, despite the importance of individual comparisons, it would be useful reporting average values and relevant correlations in a figure.

The extended reports on VO2max are not associated with the results of the study (not related to HRV values in your data) and there are not connected with the discussion section. Moreover, the changes reported on HRV are not connected to the actual internal load of the athletes. Despite you have adequately controlled the external load (based on % of VT2 and max aerobic Power) you have not inserted any comments concerning possible differences in the internal load of training in P1 and P2. Just because of these changes, you may have observed differences between your participants and not because of the training intensity per se in P2. You need to report these as limitations of the study. It is fine that you have choose to report and compare individual values, however, it would be interesting to report the average trends in a figure and correlations with all participants together.

Line 31-33. This is not a clear comment

Line35. Own studies is not an appropriate expression.

Lines 29-49. A great part is devoted to VO2max. However using HRV is normally used for other purposes, i.e. to detect training load changes. Why you focus only on VO2max? (you have no comments on this index in no other part of the paper)

L124. How was the general feeling assessed?

L125. Why 100%? Is there any data to support this?

L139. Where you base these numbers 0.19 and 0.28W? How you justify this?

L146. Briefly explain the method.

L260. Rephrase “but no or the opposite”

Reviewer 3 Report

The paper requires extensive English language revision - I am happy to review again once this has been fixed.

Abstract

It would be beneficial to include the sample size and basic participant demographics within the abstract.

Introduction

Consisting of only two (long) paragraphs, the introduction is quite dense.  Additionally, the authors do a poor job of connecting the introduction with the results presented in the discussion. While the introduction mentions VO2max and HRV, this does not come across well within the results and discussion.

I’m also not sure why the authors emphasize training intensification, would it not be clearly to the intended audience to talk about training overload, although perhaps this is an individual preference for me as a reader.

Lines 31-33: This sentence does not make sense, I am unsure what the authors are trying to convey.

Line 35: Please rephrase “own studies”, as this is not a common expression.

Lines 39-40: Please rephrase, what is here 97 of?

Lines 38-41: I do not see the relevance of discussing DNA changes as a means to predict changes in CRF – this seems out of place and in my opinion does not add value to the introduction.

Lines 29-49: There seems to be a disproportionate focus on VO2max when it is discussed throughout the remainder of the manuscript. I would spend more time introducing HRV and what it has been used for within athletic populations as this seems to be the focus of the paper.

Methods

It would be beneficial to report participant characteristics (age, height, weight) and competitive status.

I would encourage the authors to remove LF/HF indices from the manuscript as somewhat recent (Billman 2013b) research has conclusively shown the LF/HF ratio does not provide an accurate reflection of the sympathovagal balance. For ease of interpretation and reporting, I would encourage the authors to remove this measure.

Line 124: How was the participants general feeling assessed? This measure and statement are quite vague.

While the training periods are well described, the subsequent explanations offer no validity as the real load was not expressed. It would be desirable to know and report the real (actual) load completed by each subject.

Lines 154-159: What filter level and level of artifact correction was used for Kubios analysis?

Lines 167-168: Validity and reliability citations for Polar Flow should be included.

Results

It would be nice to report some form of Effect size statistic.

Try to avoid obvious statements – you would hope and expect training volume would be significantly higher in HIT compared to LIT…

Discussion

The discussion needs an extensive review and rewrite. The results of the study should be clearly described and based from the hypotheses initially developed.

The paragraphs are long, complex and often poorly structured, and the results are not discussed in an organized manner.

Author Response

(Please find attachment)

The paper requires extensive English language revision - I am happy to review again once this has been fixed.

- We will ask the Editors of the Journal for English language revision, after reading the instructions for authors, I know that it is possible.

We worked with one translator for several years, then we did not receive any linguistic comments, but this year he ended its activity. The current manuscript was translated by a translator recommended by the Editor-in-Chief of the English-language periodical published by our University. Unfortunately, as indicated by the comments of two Reviewers, the quality of the translation is poor, so I don't want to use our translator's services again.

Abstract

It would be beneficial to include the sample size and basic participant demographics within the abstract.

- As suggested, I added a short (due to the word limit in the abstract) information indicating that 7 Polish cyclists were tested.

Introduction

Consisting of only two (long) paragraphs, the introduction is quite dense.  Additionally, the authors do a poor job of connecting the introduction with the results presented in the discussion. While the introduction mentions VO2max and HRV, this does not come across well within the results and discussion.

I’m also not sure why the authors emphasize training intensification, would it not be clearly to the intended audience to talk about training overload, although perhaps this is an individual preference for me as a reader.

- As rightly suggested by all Reviewers, Introduction has been significantly changed. In the current version of the Introduction, we describe the issue of training intensification, different types of HIT training and the use of HRV measurement in the training process.

However, we do not describe the issue of overload, because we do not lead to it during the training process carried out and presented.

Lines 31-33: This sentence does not make sense, I am unsure what the authors are trying to convey.

- This sentence has been deleted in the current version of the Introduction.

Line 35: Please rephrase “own studies”, as this is not a common expression.

- This sentence has also been deleted.

Lines 39-40: Please rephrase, what is here 97 of?

- This sentence has been deleted.

Lines 38-41: I do not see the relevance of discussing DNA changes as a means to predict changes in CRF – this seems out of place and in my opinion does not add value to the introduction.

- Indeed, this fragment is unnecessary, it has been deleted.

Lines 29-49: There seems to be a disproportionate focus on VO2max when it is discussed throughout the remainder of the manuscript. I would spend more time introducing HRV and what it has been used for within athletic populations as this seems to be the focus of the paper.

- The indicated paragraph of the introduction has been changed, as suggested by all Reviewers.

Methods

It would be beneficial to report participant characteristics (age, height, weight) and competitive status.

- Information about the competitive status was added to Table 1, which characterizes the participants of the study. Age, height and weight are presented in Table 1, in the 4th, 5th and 6th row of the table. Now, they have been moved to the top of the table to make them more visible.

I would encourage the authors to remove LF/HF indices from the manuscript as somewhat recent (Billman 2013b) research has conclusively shown the LF/HF ratio does not provide an accurate reflection of the sympathovagal balance. For ease of interpretation and reporting, I would encourage the authors to remove this measure.

- As suggested, the LF/HF indicator has been removed.

Line 124: How was the participants general feeling assessed? This measure and statement are quite vague.

- The general feeling was assessed on the basis of conversations with the cyclists. Explanatory information was added in the indicated part of the manuscript.

While the training periods are well described, the subsequent explanations offer no validity as the real load was not expressed. It would be desirable to know and report the real (actual) load completed by each subject.

- The workload for each athlete is shown in the current version of the manuscript. This information has been added to the Materials and Methods section in Table 2.

Lines 154-159: What filter level and level of artifact correction was used for Kubios analysis?

- A low artifact correction threshold was used when performing the analysis. This information has been added to the indicated section of the manuscript.

Lines 167-168: Validity and reliability citations for Polar Flow should be included.

- Validity and reliability information was supported by the references and presented in the indicated section of the manuscript.

Results

It would be nice to report some form of Effect size statistic.

- The value of D-Cohen's statistic has been added to the Results section.

Try to avoid obvious statements – you would hope and expect training volume would be significantly higher in HIT compared to LIT…

- Thank you for your valuable comment. The first sentence in the Results section has been removed.

Discussion

The discussion needs an extensive review and rewrite. The results of the study should be clearly described and based from the hypotheses initially developed.

The paragraphs are long, complex and often poorly structured, and the results are not discussed in an organized manner.

- As suggested in the Discussion section, changes have been made, I hope they will be sufficient.

Round 2

Reviewer 1 Report

Agreeing with the changes made, I would only like to add a substantial improvement:

Figures 2 and 3 have multiple parts. Each figure with multiple parts should have alphabetical (e.g. A, B, C) labels on each part and all parts of each single figure should be submitted together in one file.

On the other hand, the legend of each figure should have all the necessary information to be able to interpret it.

Author Response

Agreeing with the changes made, I would only like to add a substantial improvement:

Figures 2 and 3 have multiple parts. Each figure with multiple parts should have alphabetical (e.g. A, B, C) labels on each part and all parts of each single figure should be submitted together in one file.

On the other hand, the legend of each figure should have all the necessary information to be able to interpret it.

- Thank you very much for your valuable comments. As suggested, Figures 2 and 3 have been corrected and fully described.